# Ammonium–Amine Co-Activation: Promoting the Sulfurization of Azurite and Its Effect on Xanthate Adsorption

**DOI:** 10.3390/molecules28217376

**Published:** 2023-10-31

**Authors:** Chao Su, Dianwen Liu, Jinpeng Cai, Peilun Shen

**Affiliations:** State Key Laboratory of Complex Nonferrous Metal Resources Clean Utilization, Yunnan Key Laboratory of Green Separation and Enrichment of Strategic Metal Resources, Faculty of Land Resources Engineering, Kunming University of Science and Technology, Kunming 650093, China

**Keywords:** azurite, sulfurization flotation, ammonium–amine co-activation, sulfurized products, microstructure

## Abstract

Enhanced sulfurization has always been the focus of research on the flotation of copper oxide minerals. In this study, combined ammonium–amine salts were innovatively applied to improve the sulfurization of azurite. Flotation tests were carried out to evaluate the promoting effect of ammonium–amine co-activation on the sulfurization–xanthate flotation of azurite, and the microstructure evolution of sulfurized products was investigated to reveal the mechanism underlying this promoting effect. Compared with single ammonium (amine) salt activation, ammonium–amine co-activation improved the floatability of azurite to a greater extent, i.e., the flotation recovery increased by over 4 percentage points. ToF-SIMS, ICP-OES, FESEM-EDS, AFM, XRD, and UV-vis analyses indicated that ammonium–amine co-activation combined the advantages of inorganic ammonium for buffering pH and organic amine for copper ion complexation, thus promoting the growth of sulfurized crystal products (covellite) and enhancing the adhesion stability of sulfurized products on azurite. Therefore, increasing amounts of copper sulfide components were generated under the ammonium–amine–Na_2_S system, promoting the adsorption of additional xanthate on azurite. This study provides theoretical support for the application of combined ammonium–amine salts for the sulfurization flotation of copper oxide.

## 1. Introduction

Copper is widely used in electric power systems, light industry, machinery, construction, and other industries because of its good electrical conductivity, thermal conductivity, corrosion resistance, and mechanical properties [1,2]. The raw materials for producing refined copper mainly come from copper sulfide [3]. In addition to chalcopyrite, chalcocite, bornite, and other copper sulfide minerals, copper oxide minerals are important copper raw materials. Common copper oxide minerals include malachite, azurite, chrysocolla, cuprite, tenorite, chalcanthite, and brochantite, among which malachite and azurite are the most valuable for industrial applications [4,5]. Compared with copper sulfide minerals, copper oxide minerals have the common characteristics of high solubility and poor floatability, which make it difficult to recover copper oxide minerals by direct flotation with xanthate [4,6]. Therefore, the highly efficient recovery of copper oxide has always been an important research direction for mineral workers.

The recovery processes of copper oxide can be approximately divided into three categories according to their characteristics: leaching, direct flotation, and sulfurization flotation. Acid leaching is an effective method to treat refractory copper oxide; however, it is only suitable for the treatment of copper oxide ores dominated by acidic gangue because the dissolution characteristics of carbonate gangue minerals (calcite and dolomite) are similar to those of copper oxide minerals in acidic solutions [7,8]. Although ammonia leaching can avoid the disadvantages of acid leaching, it faces some problems, such as low leaching efficiency and environmental pollution [9]. Direct flotation is the earliest reported method to deal with copper oxide minerals; the main feature of this method is that it does not need any activator for flotation but only requires collectors such as fatty acids, amines, and chelates [10,11,12,13]. Owing to the low selectivity and high cost of collectors, the large-scale application of direct flotation in copper oxide ore dressing is greatly limited. Currently, sulfurization flotation is the main method to treat copper oxide in the industry. Copper oxide mineral is pre-sulfurized with sodium sulfide and other sulfurizing reagents to produce a film of copper sulfide compounds with strong hydrophobicity on the mineral surface. Subsequently, xanthate is used to collect copper [14,15,16]. The advantages of this method are simple operation and high recovery, which are suitable for carbonate-type copper oxide ore.

The quality of sulfurization is the key to the flotation of copper oxide minerals. In the 1980s, Castro et al. studied the sulfurization flotation of malachite, chrysocolla, and tenorite and showed that the sulfurization of copper oxide minerals involved the following three parallel processes: adsorption of sulfur ions on the surface of copper oxide minerals to form copper sulfide, oxidation of sulfur ions, and desorption of oxidation products [17,18]. Raghavan et al.’s research on chrysocolla further indicated that sulfurization was not limited to the mineral surface but occurred deep into the bulk phase to form copper sulfide films [19]. At the end of the 20th century, Zhou and Chander et al. studied the sulfurization kinetics of malachite and found that discrete crystals of an independent phase were formed on the surface of malachite after sulfurization, accompanied by the formation of colloidal copper sulfide in the flotation solution [20]. The academic idea of ‘enhanced sulfurization’ has been put forward to reduce the negative effects of colloidal copper sulfide on the flotation of copper oxide minerals. Ammonium (amine) salt has shown the best application effect in industrial production in the ‘enhanced sulfurization’ of copper oxide [21,22]. According to density functional theory calculations, Wu et al. indicated that the Cu 3d orbital peak on the surface of malachite (−201) after activation by ammonium ions is closer to the Fermi level; moreover, it has high reactivity and thus is likely to interact with hydrosulfide ions (HS^−^) [23]. Feng et al. studied the effect of ethylenediamine on the sulfurization flotation of malachite and showed that pretreatment with ethylenediamine increased the number of active copper sites on the surface of malachite, which was conducive to the subsequent adsorption of sulfur ions and collectors [24]. Zuo et al. revealed that the adsorption energy of HS^−^ on tenorite was significantly higher than that of HS^−^ in the presence of NH_3_ [25].

The previous understanding that ammonium (amine) enhances the sulfurization of copper oxide minerals is based on the viewpoint of chemical adsorption. Recent studies have shown that the sulfurization of copper oxide minerals is a chemical reaction that involves two phases: copper sulfide product nucleation and crystal growth [2,26]. Liu et al. compared the surface morphology of sulfurized malachite in the absence or presence of ammonium salt and found that large amounts of sulfurized products with good crystallinity were generated on malachite in the presence of ammonium salt. The buffer effect of the double hydrolysis of NH_4_^+^ and HS^−^ on the pH of the solution is the main factor affecting the crystallization of sulfurized products [27]. Cai et al.’s research on azurite indicated that ethylenediamine can change the sulfurization path by forming ethylenediamine copper ions, thus inhibiting nucleation and promoting the stable growth of sulfurized products on azurite [28]. In view of the different regulatory characteristics of inorganic ammonium and organic amine on copper oxide minerals, combining the advantages of these two compounds may further improve the sulfurization flotation of copper oxide minerals. In this study, by considering azurite as the research object, microflotation tests were conducted to evaluate the effect of combined ammonia–amine salts on the sulfurization xanthate of azurite. On the basis of the results, the microstructure evolution of the sulfurized products under the influence of combined ammonia–amine salts was investigated to preliminarily analyze the mechanism underlying the co-activation of combined ammonia–amine salts to promote the sulfurization of azurite. This study aims to provide theoretical support for the application of combined ammonium–amine salts in the sulfurization flotation of copper oxide.

## 2. Results and Discussion

### 2.1. Floatability Studies

Figure 1a shows the functional relationship between the flotation recovery of azurite and Na_2_S concentration. When the Na_2_S concentration increased from 0 to 5 × 10^−4^ mol/L, the flotation recovery of azurite increased from 31.76% to 75.63%. With a further increase in the Na_2_S concentration, the azurite floatability decreased rapidly. When the Na_2_S concentration exceeded 1.5 × 10^−3^ mol/L, azurite had a lower flotation recovery than that in direct flotation using only xanthate and could not even be floated. This phenomenon may be closely related to the brown substance formed in the flotation pulp. Previous studies have shown that this brown substance is colloidal copper sulfide that forms during sulfurization and prevents the effective adsorption of xanthate on azurite [2,25]. Additionally, one possible reason was that overdosing Na_2_S decreased the pulp potential, which made xanthate adsorption impossible.

Figure 1b,c shows the effects of the types and concentrations of inorganic ammonium and organic amine on the flotation recovery of azurite, respectively. Results in Figure 1b show that the azurite floatability increases first and then decreases with the increase in [NH_4_^+^]. The optimal flotation recovery is obtained when the molar concentration ratio of [NH_4_^+^] to Na_2_S is 2:1 ([NH_4_^+^] = 1 × 10^−3^ mol/L). However, the impact of the accompanying anions on the azurite flotation recovery is different to some extent. According to a comprehensive evaluation, the promoting effect from high to low is (NH_4_)_2_SO_4_, (NH_4_)_2_HPO_4_, NH_4_Cl, and (NH_4_)_2_CO_3_. Looking at Figure 1c, it can be observed that when the molar concentration ratio of EDA to Na_2_S reached 0.05 ([EDA] = 2.5 × 10^−5^ mol/L), the flotation recovery of azurite increased from 75.63% to 85.68% (C_2_H_8_N_2_·H_2_SO_4_), 83.59% (C_2_H_8_N_2_·HCl), and 85.79% (C_2_H_8_N_2_·H_3_PO_4_), respectively, under the action of three organic amines. Considering the cost and the negative impact of high concentration, C_2_H_8_N_2_·H_2_SO_4_ is more suitable as a sulfurization accelerator for azurite.

Based on the difference between inorganic ammonium and organic amine in regulating the sulfurization of copper oxide [26,27], the combined ammonia–amine salts were innovatively used to improve the floatability of azurite; the results are shown in Figure 2a,b. Compared with the highest flotation recovery obtained using a single ammonium (amine) salt ((NH_4_)_2_SO_4_: 84.56% or C_2_H_8_N_2_·H_2_SO_4_: 85.68%), the combination of (NH_4_)_2_SO_4_ and C_2_H_8_N_2_·H_2_SO_4_ promoted the sulfurization–xanthate flotation of azurite to a considerable extent and increased the flotation recovery by over 4 percentage points. When the concentrations of (NH_4_)_2_SO_4_ and C_2_H_8_N_2_·H_2_SO_4_ were 4 × 10^−4^ and 2 × 10^−5^ mol/L, respectively, the flotation recovery of azurite reached 90.06%.

### 2.2. Sulfur-Containing Migration Analyses

The improved flotation recovery of azurite after the activation of combined ammonia–amine is generally related to the enhanced sulfurization surface [4]. Therefore, the distribution and composition of sulfur-containing components on the azurite surface after sulfurization were analyzed via Time-of-Flight Secondary Ion Mass Spectrometry (ToF-SIMS). Figure 3 shows the two-dimensional (2D) distribution of S^−^ and S_2_^−^ ion fragments on azurite after sulfurization. The sulfur-containing ion fragments were detected on azurite regardless of sulfurization conditions, indicating that the sulfurized products were generated on azurite after sulfurization. Compared with single Na_2_S (Figure 3a), the signal intensities of the S^−^ and S_2_^−^ ion fragments on azurite were significantly enhanced in the presence of [NH_4_^+^] (Figure 4b) or EDA (Figure 3c). This finding showed that pretreatment with ammonium (amine) salts promotes the formation of sulfurized products on azurite. Figure 3d displays that after ammonium–amine co-activation, the 2D images of the S^−^ and S_2_^−^ ion fragments on azurite after sulfurization appeared brighter, corresponding to the increased signal intensity.

The obtained S^−^ and S_2_^−^ ion fragments were normalized according to the proportion of characteristic ion fragments relative to the total number of ion fragments, as shown in Figure 4. The normalized intensities (NIs) of the S^−^ and S_2_^−^ ions fragments were 0.043 and 0.006, respectively, after treatment with single Na_2_S. The abundance of S^−^ and S_2_^−^ ions fragments on azurite increased substantially after sulfurization in the presence of [NH_4_^+^] or EDA. The NIs of the S^−^ ion fragment increased to 0.062 and 0.059, and the NIs of the S_2_^−^ ion fragment increased to 0.009 and 0.01. Under the [NH_4_^+^]/EDA-Na_2_S system, the NIs of S^−^ and S_2_^−^ ions fragments on azurite were 0.087 and 0.012, respectively, which are significantly higher than those under the single ammonium (amine) salt–Na_2_S system. These results indicate that ammonium–amine co-activation exerts a good promoting effect on the sulfurization of azurite.

Table 1 shows the residual sulfur concentration in solution versus sulfurization time under different sulfurization systems. Regardless of [NH_4_^+^] and EDA addition, the residual sulfur concentration in the solution presented a trend of first decreasing and then increasing with the increase in sulfurization time. When the sulfurization time was 5 min, the residual sulfur concentrations in the solution were relatively low at 7.59, 4.52, 4.82, and 3.26 mg/L under the four sulfurization systems: Na_2_S, [NH_4_^+^]-Na_2_S, EDA-Na_2_S, and [NH_4_^+^]/EDA-Na_2_S, respectively. An increasing amount of sulfur-containing components were transferred to the azurite surface in the presence of [NH_4_^+^] or EDA, further supporting the ToF-SIMS results. The conversion rate of sulfur from solution to surface was assessed according to the proportion of sulfur adsorption capacity to the initial sulfur concentration (16 mg/L). The highest conversion rate of sulfur was recorded for the [NH_4_^+^]/EDA-Na_2_S system (79.63%), followed by the [NH_4_^+^]-Na_2_S and EDA-Na_2_S systems (71.75% and 69.81%, respectively), and the lowest was noted for the single Na_2_S system (52.56%).

The residual sulfur concentration in the solution gradually increased when the sulfurization time exceeded 15 min. This phenomenon is widespread in the sulfurization of copper, lead, and zinc oxide ores and is attributed to the decay of sulfurized products [2,29,30]. Compared with that in the single Na_2_S system, the decay of sulfurized products was evidently weakened in the [NH_4_^+^]-Na_2_S and EDA-Na_2_S systems, especially in the [NH_4_^+^]/EDA-Na_2_S system. These results indicate that the adhesion stability of sulfurized products on the base azurite is enhanced in the presence of [NH_4_^+^] or EDA, which may be responsible for the improved sulfurization efficiency.

### 2.3. Surface Morphology Characterizations

Field Emission Scanning Electron Microscopy (FESEM) Atomic Force Microscopy (AFM) were used to characterize the difference in surface morphology to analyze the microscopic performance of improved sulfurization efficiency and adhesion stability of sulfurized products on azurite in the [NH_4_^+^]/EDA-Na_2_S system. Figure 5 shows the FESEM surface morphology of azurite treated under different sulfurization conditions. The azurite surface was covered with a layer of products after sulfurization, appearing as both a granular product and a flake product with a regular shape. These observations are consistent with the findings of previous studies, according to which the sulfurization of copper oxide minerals is a chemical reaction involving crystallization. Compared with those under single Na_2_S, increasing amounts of sulfurized products were formed on azurite in the presence of [NH_4_^+^] or EDA. Energy-Dispersive X-ray Spectroscopy (EDS) surface scan mapping of element S in Figure 6 shows that the S mass concentrations in the azurite treated with the [NH_4_^+^]-Na_2_S and EDA-Na_2_S systems were 1.2% and 1.1%, respectively, which are higher than that in the azurite treated with Na_2_S alone (0.8%). Additionally, the generated flake product exhibited a large radial size in the presence of [NH_4_^+^] or EDA. Therefore, pretreatment with [NH_4_^+^] or EDA promotes the growth of sulfurized crystal products, indicating that nucleation is reduced at the initial stage of sulfurization after [NH_4_^+^] or EDA is introduced. Figure 5d and Figure 6d further confirm the synergistic-promoting effect of combined ammonium–amine salts on the sulfurization of azurite. In the [NH_4_^+^]/EDA-Na_2_S system, the S mass concentration in azurite reached 1.4%, and the radial size of sulfurized products increased further. The radial dimensions of the three flake products were 108, 122, and 131 nm.

Figure 7 shows the AFM surface morphology of azurite treated with different sulfurization conditions. The azurite surface was covered with a layer of columnar protrusion structure. Combined with the FESEM test results, this figure reveals that the columnar protrusion structure is a newly generated sulfurized product. From the FESEM observations, the geometric size of columnar protrusions on azurite increased significantly in the presence of [NH_4_^+^] or EDA. Owing to the high spatial structure, the corresponding root-mean-square roughness (Rq) values of azurite were 28.0 (Figure 7b) and 29.3 (Figure 7c), which are higher than those of azurite treated with single Na_2_S (Figure 7a, 22.1). After pretreatment with combined [NH_4_^+^] and EDA, the geometric size and spatial height of the generated columnar protrusions were significantly larger than those of single [NH_4_^+^] or EDA, and the Rq was 35.2. These results support the conclusion from FESEM that ammonia–amine co-activation promotes the growth of sulfurized crystal products.

### 2.4. Product Composition Analyses

X-ray Diffraction (XRD) testing was performed to analyze the effect of [NH_4_^+^] or EDA on the composition and crystallinity of sulfurized products. The characteristic peak of sulfurized products could hardly be found on the XRD pattern of the sulfurized copper oxide minerals because the content of the generated sulfurized products is lower than the limit of XRD detection [2,28]. Therefore, the sulfurized azurite was selectively dissolved by dilute sulfuric acid to obtain the sulfurized products for XRD detection, and the results of the purified products are shown in Figure 8a. Four diffraction peaks appeared in the range of 20° to 70° at around 29.3°, 31.8°, 48.0°, and 59.3°, which were well matched with those of covellite (syn-CuS). The copper-deficient phase (Cu_2−x_S, 0 < x ≤ 1) is more thermodynamically stable than the copper-rich phase (Cu_2_S), which is in agreement with our findings. Additionally, the diffraction pattern of the sulfurized product showed the following three characteristics: (i) the base line of the diffraction peak rose in a small diffraction angle range of 5°–20°; (ii) the intensity of the diffraction peak was low; and (iii) the diffraction peaks broadened. These results show that the sulfurized products exist as fine grains, which is consistent with the morphology of sulfurized products observed by FESEM.

Figure 8b–d shows the XRD patterns of sulfurized products of azurite in the presence of [NH_4_^+^] or EDA. The sulfurized products remained covellite after the addition of [NH_4_^+^] or EDA prior to Na_2_S, indicating that pretreatment with [NH_4_^+^] or EDA has a minimal effect on the products’ composition. Compared with that under the single Na_2_S system, new diffraction peaks located around 27.1°, 27.7°, 32.8°, and 52.7° and corresponding to (100), (101), (006) and (108) crystal faces of covellite, respectively, were detected in the XRD patterns of sulfurized products in the presence of [NH_4_^+^] or EDA. The diffraction peaks in the XRD patterns of sulfurized products after adding [NH_4_^+^] or EDA prior to Na_2_S also had less broadening and more strength than those under the single Na_2_S system, and these phenomena were highly pronounced in the [NH_4_^+^]/EDA-Na_2_S system. According to Scherrer’s equation (D = Kλ/(βcosθ)), the grain size (D) is inversely proportional to the half-width (β), and a small half-peak width corresponds to a large grain size. Therefore, the sulfurized product covellite with good crystallinity was formed on azurite in the presence of [NH_4_^+^] or EDA, as observed through FESEM and AFM surface morphology.

### 2.5. Xanthate Adsorption Studies

Figure 9 shows the adsorption capacity of sodium butyl xanthate (SBX) on azurite treated with different sulfurization conditions. The adsorption capacity of SBX on the surface of sulfurized azurite increased with the initial dosage, regardless of [NH_4_^+^] or EDA addition. Compared with that under single Na_2_S, the adsorption capacity of SBX on sulfurized azurite increased in the presence of [NH_4_^+^] or EDA due to the increasing amount of copper sulfide components that promote the effective adsorption of xanthate on azurite. Moreover, xanthate showed the largest adsorption capacity on azurite in the [NH_4_^+^]/EDA-Na_2_S system, which is consistent with the flotation test results.

### 2.6. Mechanism of Ammonium–Amine Co-Activation in Promoting Sulfurization

Flotation tests and surface characterization proved that the combined ammonium–amine salts are an excellent activator for the sulfurization flotation of azurite. The increasing adsorption capacity of xanthate leads to the improved floatability of azurite in the [NH_4_^+^]/EDA-Na_2_S system, which is closely related to the formation of additional sulfurized products on azurite after [NH_4_^+^]/EDA addition prior to Na_2_S. As shown in Equation (1), Cu(OH)_2_(aq) and HS^-^ are the main initial reactants that provide a copper source and sulfur source to produce copper sulfide in the absence of [NH_4_^+^]/EDA [28]. Combined with previous studies, the present work suggests that ammonium–amine co-activation fully combines the advantages of inorganic ammonium and organic amine to regulate the sulfurization of azurite from two different aspects. The double hydrolysis between NH_4_^+^ and HS^−^ buffers the pH of the flotation solution, which changes the occurrence state of the sulfur source (Equation (2)). Owing to the strong complexation ability of EDA with copper ions, sulfurization must undergo the intermediate state of the complex from the initial state to the product state (Equation (3). Therefore, in the [NH_4_^+^]/EDA-Na_2_S system, in addition to reaction 1, other pathways to produce copper sulfide may also exist, like Equations (4)–(6). The change in the sulfurization reaction pathway slowed down the nucleation rate of the copper sulfide product in the initial stage of the sulfurization, thus providing a good pulp environment for the stable growth of the copper sulfide product on the base azurite and thus enhancing the adhesion stability of the copper sulfide product on azurite. As a result, an increased amount of copper sulfide components were generated on azurite, which promoted the effective adsorption of xanthate on azurite. The hypothesised mechanism model is shown in Figure 10.
(1)$$Cu{\left(OH\right)}_{2}+HS^{-}\to CuS+H_{2}O+OH^{-}$$
(2)$$2NH_{4}{}^{+}+2HS^{-}\to 2NH_{3}+H_{2}S(aq)+H_{2}S(g)$$
(3)$$Cu{\left(OH\right)}_{2}+2EDA\to Cu{\left(EDA\right)}_{2}{}^{2+}+2OH^{-}$$
(4)$$C{u(OH)}_{2}+H_{2}S\to CuS+2H_{2}O$$
(5)$$Cu{\left(EDA\right)}_{2}{}^{2+}+HS^{-}+OH^{-}\to CuS+2EDA+H_{2}O$$
(6)$$Cu{\left(EDA\right)}_{2}{}^{2+}+H_{2}S+2OH\to CuS+2EDA+2H_{2}O$$

## 3. Materials and Methods

### 3.1. Materials

Azurite samples were obtained from Yangchun, Guangdong Province, China. The collected azurite samples were cleaned via ultrasonic processing to remove slime and then dried for crushing, impurity removal, grinding, and screening to obtain samples in a particle size range of −74 + 37 μm for the flotation test. Figure 11 shows the XRD pattern of the azurite sample. Except for a few impurity peaks, the main diffraction peak location and peak intensity are consistent with the standard pattern of azurite. The chemical analysis results presented in Table 2 show that the azurite sample contains 53.85% Cu, indicating that the purity of this sample is >97%. The detailed parameters of the flotation reagents used in this study are shown in Table 3.

### 3.2. Methods

#### 3.2.1. Flotation Tests

The flotation tests were conducted in a self-made Hallimond tube. Azurite samples weighing 0.5 g were transferred to a beaker with certain deionized water and then added with ammonium ([NH_4_^+^]) or ethylenediamine (EDA) salt, sodium sulfide, and the collector to adjust the slurry for 3, 5, and 3 min, respectively. The pulp pH was adjusted to 9.5 ± 0.1 before the collector was added. After the reaction was completed, the slurry was transferred into a Hallimond tube with a volume of 50 mL for flotation for 8 min at an aeration rate of 40 mL/min. The flotation recovery of azurite was calculated based on the weight percentage of floating products to total products. Each test was repeated at least three times, and the average value was used to assess azurite floatability.

#### 3.2.2. ToF-SIMS Analysis

TOF-SIMS (ION-TOF, Munster, Germany) was used to analyze the distribution and composition information of sulfur-containing components on azurite after sulfurization. The ion source was Bi^3+^, the energy was 30 kV, the analysis area was 500 µm × 500 µm, and the surface analysis time was 66 s. The positive spectra of secondary ions were corrected by CH_3_^+^, Na^+^, and Cu^+^, and the negative spectra were corrected by CH^−^, Cl^−^, and Cu^−^. The four azurite samples used for testing were treated with the following: sample a, 5 × 10^−4^ mol/L Na_2_S; sample b, 5 × 10^−4^ mol/L (NH_4_)_2_SO_4_ and 5 × 10^−4^ mol/L Na_2_S; sample c, 2.5 × 10^−5^ mol/L C_2_H_8_N_2_·H_2_SO_4_ and 5 × 10^−4^ mol/L Na_2_S; and sample d, 4 × 10^−4^ mol/L (NH_4_)_2_SO_4_, 2 × 10^−4^ mol/L C_2_H_8_N_2_·H_2_SO_4_ and 5 × 10^−4^ mol/L Na_2_S. The reaction time of the reagents was consistent with the flotation tests. The reaction samples were dried in a vacuum oven and then divided into two parts, one for TOF-SIMS detection and the other for FESEM-EDS detection.

#### 3.2.3. Inductively Coupled Plasma-Optical Emission Spectrometry (ICP-OES) Analysis

ICP-OES (PlasmaQuant PQ9000, Jena, Germany) was used to analyze the residual sulfur concentration in the solution after sulfurization. Azurite samples of 0.5 g were treated with ammonium (amine) salts for 3 min and then added with 5 × 10^−4^ mol/L Na_2_S for conditioning at different times (1, 3, 5, 7, 10, 15, 30, and 60 min). The dosage of ammonium (amine) salts was based on the sulfurization conditions. After reaction, 10 mL of supernatant was extracted and transferred to the centrifuge tube, which was then centrifuged for 1 h at 4000 r/min. After centrifugation, 3 mL of supernatant was collected for testing.

#### 3.2.4. FESEM-EDS and AFM Analyses

FESEM (FEI Nova Nano SEM 450, Hillsboro, OR, USA) and AFM (Bruker Dimension ICON, Münster, Germany) were performed to analyze the surface morphology of the sulfurized azurite, and EDS (Genesis 2000, EMETEK, Mahwah, NJ, USA) was used to obtain the scan mapping of S on the azurite surface. The azurite samples were sprayed with platinum before FESEM-EDS testing to increase their surface conductivity. During the testing, the acceleration was set to 15 kV, the AFM analysis area was 5 μm × 5 μm, and the scanning frequency was 1 Hz. After the testing, NanoScope Analysis 1.7 software was used to process the obtained data, including the formation of intuitive 3D profiles and surface roughness statistics. The sampling for AFM testing was the same as that for FESEM testing, except that the samples in AFM testing were rectangular pieces with an approximate dimension of 5 mm × 5 mm × 2 mm.

#### 3.2.5. XRD Analysis

XRD (X’Pert Pro, Malvern Panalytical Ltd., Malvern, Worcestershire, UK) was performed to analyze the composition and crystallinity of sulfurized products. Cu-Kα ray (λ = 0.15405 nm) was used as an X-ray excitation source, and the acceleration voltage and acceleration current were 40 kV and 40 mA, respectively. The scanning speed was 3°/min in the range of 5° to 90°. After testing, Jade 6.0 software was used to analyze the obtained pattern.

#### 3.2.6. Ultraviolet–Visible Spectroscopy (UV-Vis) Measurement

UV-vis spectroscopy (UV-2700, Shimadzu, Japan) was used to determine the adsorption of SBX on the sulfurized azurite. In accordance with the sulfurization conditions, azurite samples of 0.5 g were successively treated with ammonium (amine) salts, Na_2_S, and SBX of different concentrations (5, 10, 15, and 20 mg/L) for 3, 5, and 3 min, respectively. The pulp pH was adjusted to 9.5 ± 0.1 before adding SBX. After the reaction, 10 mL of supernatant was extracted and transferred to the centrifuge tube, which was then centrifuged for 1 h at 4000 r/min. After centrifugation, 3 mL of supernatant was collected for testing.

## 4. Conclusions

In this study, combined ammonium–amine salts were innovatively applied to improve the sulfurization–xanthate flotation of azurite. The microstructure evolution of the sulfurized products under the action of combined ammonia–amine salts was also systematically investigated. The main conclusions are as follows:
The combined ammonium–amine salts are an excellent activator for the sulfurization flotation of azurite. Compared with single ammonium (amine) salts, the combined ammonium–amine salts can improve the floatability of azurite to a greater extent and increase the flotation recovery by over 4 percentage points.Under the [NH_4_^+^]/EDA-Na_2_S system, an increased amount of copper sulfide components are generated on azurite. This phenomenon is closely related to improved adhesion stability.Pretreatment with [NH_4_^+^]/EDA does not change the phase composition of sulfurized products, which mainly exist in the form of covellite (syn-CuS), but substantially promotes the growth of a covellite crystal.The abundant copper sulfide components promote the adsorption of additional xanthate on azurite in the [NH_4_^+^]/EDA-Na_2_S system.

## Figures and Tables

**Figure 1 molecules-28-07376-f001:**
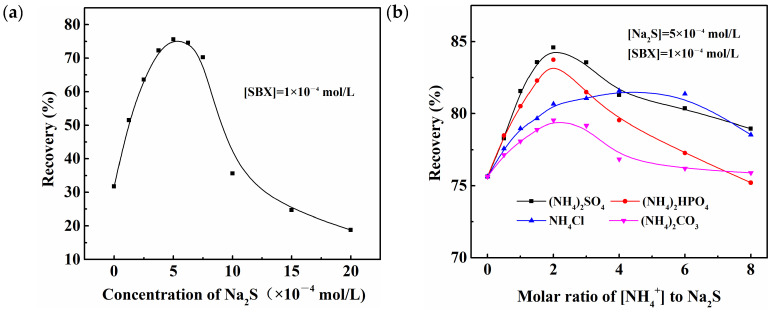
Flotation recoveries of azurite as a function of (**a**) Na_2_S concentration, (**b**) molar ratio of [NH_4_^+^] to Na_2_S, and (**c**) molar ratio of EDA to Na_2_S.

**Figure 2 molecules-28-07376-f002:**
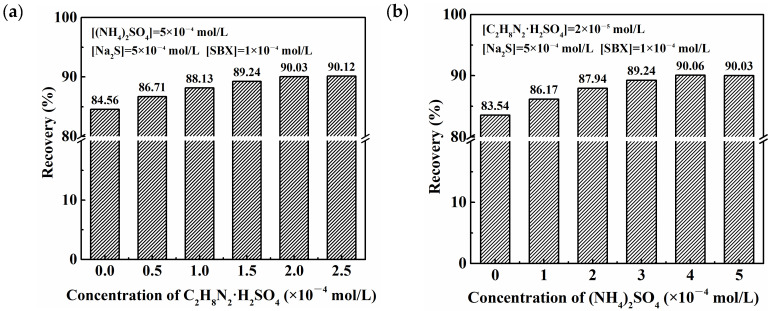
Flotation recoveries of azurite as a function of (**a**) C_2_H_8_N_2_·H_2_SO_4_ concentration and (**b**) (NH_4_)_2_SO_4_ concentration.

**Figure 3 molecules-28-07376-f003:**
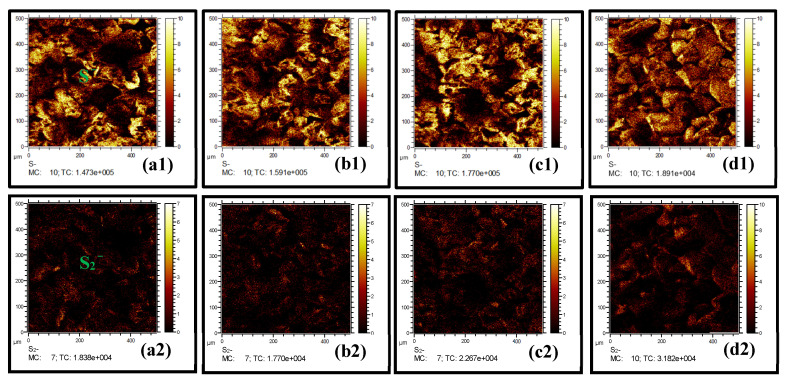
Two-Dimensional distribution of S^−^ and S_2_^−^ ion fragments on azurite treated with (**a**) Na_2_S, (**b**) [NH_4_^+^]-Na_2_S, (**c**) EDA-Na_2_S, and (**d**) [NH_4_^+^]/EDA-Na_2_S.

**Figure 4 molecules-28-07376-f004:**
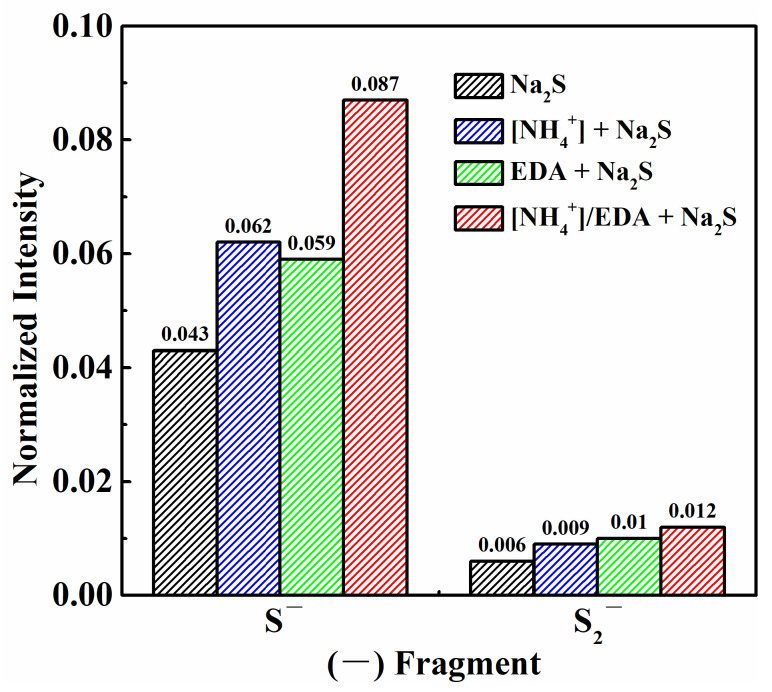
Statistical analysis results of NI of S^−^ and S_2_^−^ ion fragments on azurite.

**Figure 5 molecules-28-07376-f005:**
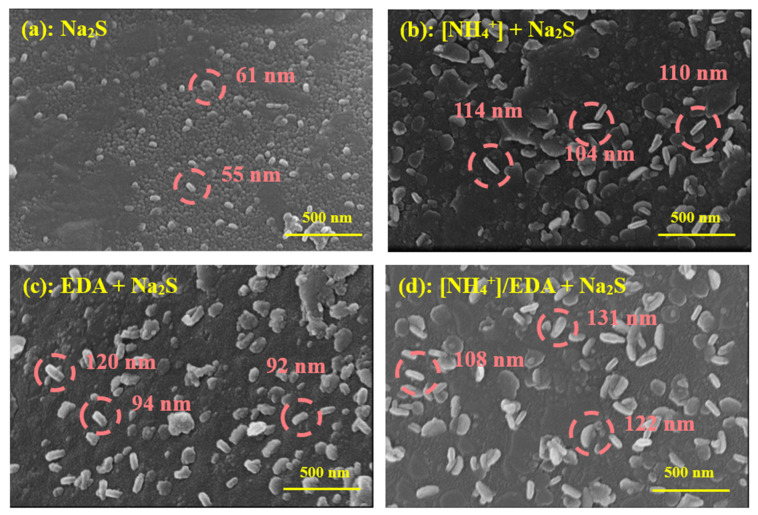
FESEM surface morphology of azurite treated with (**a**) Na_2_S, (**b**) [NH_4_^+^] + Na_2_S, (**c**) EDA + Na_2_S, and (**d**) [NH_4_^+^]/EDA + Na_2_S under 200,000× magnification.

**Figure 6 molecules-28-07376-f006:**
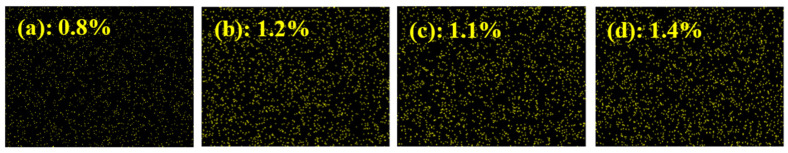
EDS surface scan mapping of element S of azurite treated with (**a**) Na_2_S, (**b**) [NH_4_^+^] + Na_2_S, (**c**) EDA + Na_2_S, and (**d**) [NH_4_^+^]/EDA + Na_2_S.

**Figure 7 molecules-28-07376-f007:**
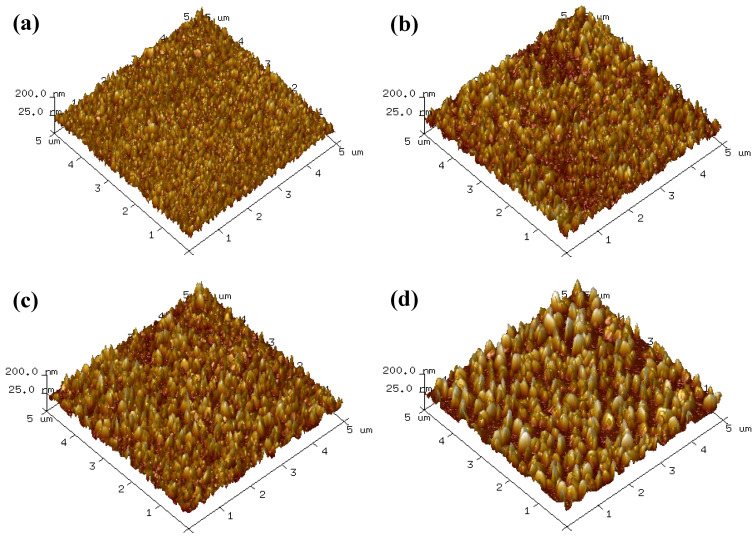
AFM surface morphology of azurite treated with (**a**) Na_2_S, (**b**) [NH_4_^+^] + Na_2_S, (**c**) EDA + Na_2_S, and (**d**) [NH_4_^+^]/EDA + Na_2_S.

**Figure 8 molecules-28-07376-f008:**
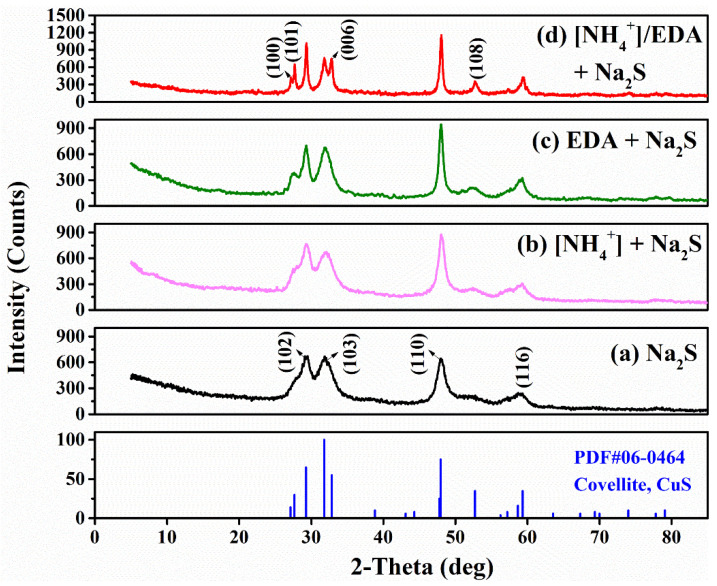
XRD pattern of sulfurized products of azurite treated with (**a**) Na_2_S, (**b**) [NH_4_^+^] + Na_2_S, (**c**) EDA + Na_2_S, and (**d**) [NH_4_^+^]/EDA + Na_2_S.

**Figure 9 molecules-28-07376-f009:**
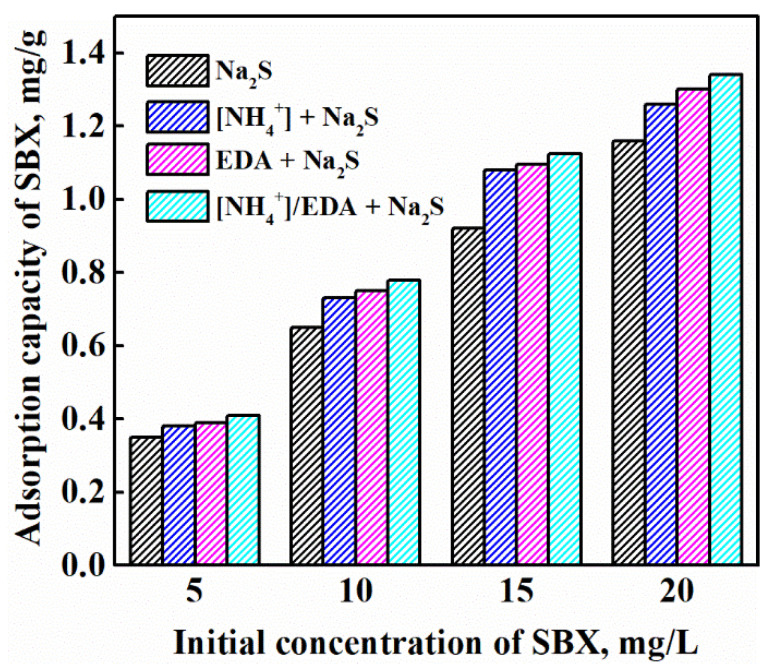
Adsorption capacity of SBX on sulfurized azurite.

**Figure 10 molecules-28-07376-f010:**
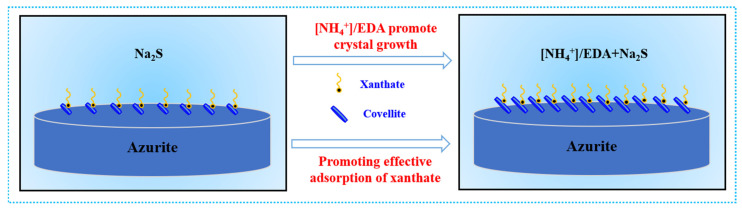
Hypothesized mechanism model of [NH_4_^+^]/EDA co-activation promoting the sulfurization–xanthate flotation of azurite.

**Figure 11 molecules-28-07376-f011:**
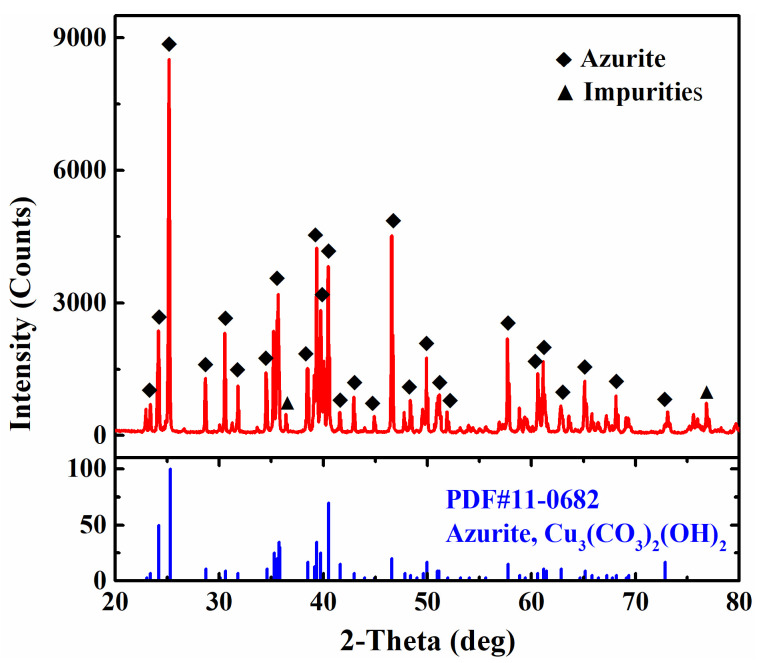
XRD pattern of the azurite sample.

**Table 1 molecules-28-07376-t001:** Residual sulfur concentration in solution versus sulfurization time (mg/L).

Sulfurization Time (Min)	Sulfurization System			
	Na_2_S	[NH_4_^+^]-Na_2_S	EDA-Na_2_S	[NH_4_^+^]/EDA-Na_2_S
1	13.27	11.80	11.32	10.54
3	9.07	6.55	5.07	4.31
5	7.59	4.52	4.83	3.26
7	7.66	4.58	4.81	3.27
10	7.89	4.84	4.85	3.29
15	8.09	4.87	4.86	3.29
30	8.64	4.90	4.91	3.30
60	9.55	5.32	5.26	3.45

**Table 2 molecules-28-07376-t002:** Chemical analysis result of the azurite sample.

Element.	Cu	Fe	CaO	Al_2_O_3_	MgO	SiO_2_
Content (%)	53.85	0.16	0.28	0.50	0.18	0.88

**Table 3 molecules-28-07376-t003:** Detailed parameters of flotation reagents.

Reagent	Chemical Formula	Purity	Function	Company
Sulfuric	H_2_SO_4_	98%	Purifying sulfurized products	Tiefeng
Sodium hydroxide	NaOH	99%	pH regulator	Aladdin
Sodium sulfide	Na_2_S·9H_2_O	99%	Activator	Aladdin
Ammonium sulfate	(NH_4_)_2_SO_4_	99%	Regulator	Aladdin
Diammonium hydrogen phosphate	(NH_4_)_2_HPO_4_	99%	Regulator	Aladdin
ammonium chloride	NH_4_Cl	99%	Regulator	Aladdin
ammonium carbonate	(NH_4_)_2_CO_3_	99%	Regulator	Aladdin
Ethylenediamine sulfate	C_2_H_8_N_2_·H_2_SO_4_	99%	Regulator	Aladdin
Ethylenediamine hydrochloride	C_2_H_8_N_2_·2HCl	99%	Regulator	Aladdin
Ethylenediamine phosphate	C_2_H_8_N_2_·H_3_PO_4_	Industrial grade	Regulator	Tiefeng
Sodium butyl xanthate	C_4_H_9_OCSSNa	90%	Collector	Aladdin

## Data Availability

Data will be made available on request.

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
