# Peer review of "Ammonium–Amine Co-Activation: Promoting the Sulfurization of Azurite and Its Effect on Xanthate Adsorption"

_molecules, 2023, doi:10.3390/molecules28217376_

Round 1

Reviewer 1 Report

Comments and Suggestions for Authors

Dear authors!

The article is devoted to flotation of azurite with ammonium-amine co-activation. The research topic is relevant and the article would be of interest to readers. The section "introduction" is well written and contains a sufficient number of references to articles. 

Recommendations for authors:

1. The term "sulfurization" is duplicated in the title of the paper

2. The authors of the article should adjust the structure of the article. The section "materials and methods of research" should come after the introduction section, before the results and discussion. Since understanding of the results requires understanding of the object.

The authors have done an excellent job, the article shows new data on azurite flotation with ammonium-amine co-activation with justification of the mechanism.

Author Response

Thank you again for your comments concerning our manuscript entitled “Ammonium–amine co-activation promoting the sulfurization sulfurization of azurite and its effect on xanthate adsorption” (molecules-2617420). Those comments are all valuable and very helpful for revising and improving our paper, and provide the important guiding significance to our studies. We have studied the comments carefully and have made corrections, which we hope to meet with approval. This revised portion are marked in red in the paper. The main corrections in the paper and the respond to your comments are as follows:

Q1: 1. The term "sulfurization" is duplicated in the title of the paper.

Response 1: Thanks for your comments. The error has been modified.

Q2: The authors of the article should adjust the structure of the article. The section "materials and methods of research" should come after the introduction section, before the results and discussion. Since understanding of the results requires understanding of the object.

Response 2: Thanks for your comments. Based on your comment, we have rearranged the order of the parts.

I hope this revised paper will get your approval.

We deeply appreciate your review of our manuscript, and we look forward to receiving your reply.

Reviewer 2 Report

Comments and Suggestions for Authors

Review of the article molecules-2617420: Ammonium–amine co-activation promoting the sulfurization of azurite and its effect on xanthate adsorption.

The article deals with the use of NH4+/amine compounds as promoters of the sulfurization of azurite mineral through sodium sulfide for the copper concentration by the flotation process using a xanthate compound as a collector. Certainly, the experimental results (by the way there are many) confirm the proposed mechanism by the authors, however, the use of NH4+ as a promoter for the sulfurization step in the flotation of oxidized copper minerals has been previously studied (tenorite). This study only confirms the previously proposed mechanism applied to azurite as the oxidized copper mineral sample.

The article can be published after attending some comments, some are simple but others deserve attention. 

Comments:

* Figure 1. Is there any explanation for the recovery differences only by changing the anion when inorganic salts were used? Or by the functional group change in the case of the use of amines?

* In Fig. 1 appears two lines are denominated unwashed and washed. Which corresponds to the description provided in the text? Lines 104-112.

* What is the meaning of EDA? Define all abbreviations when are used for the first time in the manuscript. In the same sense, avoid using abbreviations as keywords.

* Figure 5, provide the scale lines and units for the four micrographs shown.

* Mappings presented in Fig. 6 are unconvincing since in Figs. b, c, and d is not possible to appreciate a significant difference between sulfur content. Why are not the CuS grains shown pixelated? According to the micrographs, it is evident that there is more CuS in Fig. 5d, then is more suitable to present the EDS spectrum for a single grain with the end to demonstrate the presence of CuS on the azurite surface. If the authors continue with the idea of showing the differences in the S contents on the azurite surface, I recommend taking the EDS in the specified area and showing the spectrums with the analysis of elemental content as a table.  

* In the description of Fig. 9 is mentioned an FTIR spectrum of the sulfurized azurite, however such spectrum (or description) does not appear in the manuscript.

* According to the results of xanthate adsorption (Fig. 9), there is a greater accumulation of xanthate when the sulfurization is carried out in the presence of NH4+/amine. Then, this fact should be reflected in the proposed mechanism in Fig. 10, that is, the greater the amount of CuS present, the greater the amount of xanthate adsorbed. Literally, should appear in the representation more molecules of xanthate when there is more amount of covellite (left side of Fig. 10).

* In Fig. 10, usually, the collector molecule is represented by a polar head that is adsorbed on the mineral surface and a hydrocarbon (hydrophobic) tail oriented to the solution. In its current form can cause confusion due to azurite is represented as a blue circle and the collector as a single chain.

* Really what is the effect of the presence of the nitrogenized compounds in the sulfurization process of azurite? In the manuscript is mentioned that these compounds catalyze the formation of copper sulfide compounds. Would this be understood, the ammonium ion accelerating the CuS formation reaction? In my opinion, the ammonium ion acts in the formation of stable ammonium-copper complexes, avoiding the instantaneous precipitation of the Cu2+ ion released of the azurite surface as Cu(OH)2, allowing the free reaction of the SH- ion and thus promoting the formation of CuS.

* Place the chemical reactions of the process for a better explanation of the proposed mechanism.

* In Table 3, sodium sulfide is an activator since is the compound that facilitates the adsorption of the collector in the azurite surface through the formation of superficial CuS. The nitrogenized compounds are promoters since only modify the medium and do not participate directly in the flotation process.  

* Lines 458 and 508, is mentioned the conditioning times as 3, 5, and 3 min. Is it correct? 

Author Response

Thank you again for your comments concerning our manuscript entitled “Ammonium–amine co-activation promoting the sulfurization of azurite and its effect on xanthate adsorption” (molecules-2617420). Those comments are all valuable and very helpful for revising and improving our paper, and provide the important guiding significance to our studies. We have studied the comments carefully and have made corrections, which we hope to meet with approval. This revised portion are marked in red in the paper. The main corrections in the paper and the respond to your comments are as follows:

Q1: Figure 1. Is there any explanation for the recovery differences only by changing the anion when inorganic salts were used? Or by the functional group change in the case of the use of amines?

Response 1: Thanks for your comments. The effect of anion on ammonium salt promoting the sulfurization of copper oxide has been concerned by researchers. In the sulfurization flotation of chrysocolla, the promoting effect of ammonium phosphate is significantly better than that of other ammonium salts. It is pointed out that the good promoting effect of ammonium phosphate may be related to the hydrolysis of phosphate group (https://doi.org/10.1016/j.mineng.2020.106300). However, Liu et al. indicated that compared with other ammonium salts, ammonium sulfate has better effect during the sulfurization flotation of malachite (CNKI:CDMD:1.2008.112640). Overall, there is no consensus on the effect of anions. Therefore, in this work, we only compare the effects of different ammoniums or amines salts on the sulfurization flotation of azurite. In addition, this study focuses on the role of combined ammonium-amine salts rather than anions.

Q2: In Fig. 1 appears two lines are denominated unwashed and washed. Which corresponds to the description provided in the text? Lines 104-112.

Response 2: Thanks for your comments. Based on your comment, we have removed the excess red lines.

Q3: What is the meaning of EDA? Define all abbreviations when are used for the first time in the manuscript. In the same sense, avoid using abbreviations as keywords.

Response 3: Thanks for your comments. Based on your comment, the abbreviations in our manuscript have been redefined. In the abstract and keywords, the abbreviations of ammonium/amine salt are deleted. In section 2.2.1, the abbreviations of ammonium ([NH4+]) and amine (EDA) were defined. In this manuscript, [NH4+] represents inorganic ammonium salts, and EDA represents ethylenediamine salts. As for the testing method in abstract, such as ToF-SIMS, ICP-OES, FESEM-EDS, AFM, XRD and UV-vis, we request that the abbreviation be retained due to the excessive word of the full name.

Q4: Figure 5, provide the scale lines and units for the four micrographs shown.

Response 4: Thanks for your comments. I have added the scale lines and units in FESEM surface morphology.

Q5: Mappings presented in Fig. 6 are unconvincing since in Figs. b, c, and d is not possible to appreciate a significant difference between sulfur content. Why are not the CuS grains shown pixelated? According to the micrographs, it is evident that there is more CuS in Fig. 5d, then is more suitable to present the EDS spectrum for a single grain with the end to demonstrate the presence of CuS on the azurite surface. If the authors continue with the idea of showing the differences in the S contents on the azurite surface, I recommend taking the EDS in the specified area and showing the spectrums with the analysis of elemental content as a table.

Response 5: Thanks for your comments. Although the difference between sulfur content is not very obvious, combined with the surface morphology, the good promoting effect of the combined ammonium-amine salt has been reflected. In addition, EDA surface analysis is a semi-quantitative method. Therefore, a variety of methods have been used in this work to confirm this conclusion. Especially ICP-OES with quantitative analysis ability. Regarding point scanning and surface scanning, the authors agree that the results of surface scanning are more representative. Although sometimes the results of the analysis may not be as good as point analysis.

Q6: In the description of Fig. 9 is mentioned an FTIR spectrum of the sulfurized azurite, however such spectrum (or description) does not appear in the manuscript.

Response 6: Thanks for your comments. We have corrected this error by removing the infrared spectrum in the title.

Q7: According to the results of xanthate adsorption (Fig. 9), there is a greater accumulation of xanthate when the sulfurization is carried out in the presence of NH4+/amine. Then, this fact should be reflected in the proposed mechanism in Fig. 10, that is, the greater the amount of CuS present, the greater the amount of xanthate adsorbed. Literally, should appear in the representation more molecules of xanthate when there is more amount of covellite (left side of Fig. 10).

Response 7: Yes, the results of the paper support the conclusion that more copper sulfide production promotes xanthate adsorption in the presence of NH4+/amine. The fact has been presented in hypothesised mechanism model in Fig. 11. It may be because the expression form of xanthate molecules is not correct, resulting in unintuitive results. Based on your suggestions below, we have modified Fi. 11.

Q8: In Fig. 10, usually, the collector molecule is represented by a polar head that is adsorbed on the mineral surface and a hydrocarbon (hydrophobic) tail oriented to the solution. In its current form can cause confusion due to azurite is represented as a blue circle and the collector as a single chain.

Response 8: Thanks for your comments. We have modified the hypothesised mechanism model in Fig. 11.

Q9: Really what is the effect of the presence of the nitrogenized compounds in the sulfurization process of azurite? In the manuscript is mentioned that these compounds catalyze the formation of copper sulfide compounds. Would this be understood, the ammonium ion accelerating the CuS formation reaction? In my opinion, the ammonium ion acts in the formation of stable ammonium-copper complexes, avoiding the instantaneous precipitation of the Cu2+ ion released of the azurite surface as Cu(OH)2, allowing the free reaction of the SH- ion and thus promoting the formation of CuS.

Response 9: Thanks for your comments. The occurrence state of copper components has been calculated by visual MINTEQ. In the [NH4+]/EDA-Na2S system, the relative content of copper ammonia complex ion is low. Therefore, the present work suggested that ammonium–amine co-activation fully combines the advantage of inorganic ammonium and organic amine to regulate the sulfurization of azurite from two different aspects. The double hydrolysis between NH4+ and HS− buffers the pH of flotation solution, which changes the occurrence state of the sulphur source (Eq. (2)). Owing to the strong complexation ability of EDA to copper ions, the sulfurization must undergo the intermediate state of the complex from the initial state to the product state (Eq. (3)). Therefore, in the [NH4+]/EDA-Na2S system in addition to reaction 1, other pathways to produce copper sulfide may also exist, like Eq. (4), Eq. (5) and Eq. (6). The change of the sulfurization reaction pathway slowed down the nucleation rate of the copper sulphide product in the initial stage of the sulfurization, thus providing a good pulp environment for the stable growth of the copper sulphide product on the base azurite and thus enhancing the adhesion stability of copper sulphide product on azurite. As a result, an increased amount of copper sulphide components were generated on azurite, which promoted the effective adsorption of xanthate on azurite.

               (1)

          (2)

           (3)

                   (4)

     (5)

    (6)

Q10: Place the chemical reactions of the process for a better explanation of the proposed mechanism.

Response 10: Thanks for your comments. Based on your comment, the possible sulfurization pathway has been presented in section 3.6.

Q11: In Table 3, sodium sulfide is an activator since is the compound that facilitates the adsorption of the collector in the azurite surface through the formation of superficial CuS. The nitrogenized compounds are promoters since only modify the medium and do not participate directly in the flotation process.

Response 11: Thanks for your comments. Based on your comment, we have modified the description about the sodium sulphide and ammonium/amine salt. Sodium sulphide acts activatior, and ammonium/amine salt act regulator, which mainly plays a role in regulating the sulfurization process.

Q12: Lines 458 and 508, is mentioned the conditioning times as 3, 5, and 3 min. Is it correct?

Response 12: Yes, in previous studies, the action time of ammonium salt and xanthate was generally 3min. In addition, in this study, based on ICP-OES tests, we found that 5min is the most appropriate for azurite vulcanization.

I hope this revised paper will get your approval.

We deeply appreciate your review of our manuscript, and we look forward to receiving your reply.

Reviewer 3 Report

Comments and Suggestions for Authors

In this manuscript, the authors were concerned with improving azurite flotation by sulfurization process. A major finding was that using a mixture of ammonium ions and ethylenediamine substantially promoted the sulfurization reaction at azurite surface and hence improved the xanthate adsorption and azurite flotation recovery. The introduction provided a comprehensive review of the development of sulfurization process in 1970s and its application in copper mineral flotation. The experiment procedure and methods were clearly presented. The topic was closely related to Molecules. Therefore, the reviewer suggested publishing this work after minor revisions.

The reviewer raised the following points for authors’ consideration for revision,

1.       Lines 109-111: The authors attributed the decrease in flotation recovery when overdosing Na2S to the formation of colloidal copper sulfide that prevented xanthate adsorption. However, there might be other possibilities that can cause low flotation recovery in the presence of extra amounts of Na2S. One possible reason was that overdosing Na2S decreased the potential (Eh), which made xanthate adsorption impossible. Without ruling out this possibility, the authors’ statement may be inaccurate and questionable.

2.       What is the washing procedure? What is the difference between unwashed and washed samples shown in Fig. 1a?

3.       Line 257: What is the definition of “average sizes”? Does it refer to the length of the flakes?

4.       Lines 317-319: Will the sulphuric acid washing change the surface properties of the sulfurized samples for XRD analysis?

5.       In Fig. 10, the authors used short yellow curves to represent xanthate molecules. It would be better if the authors can use a structure that shows the hydrophilic head and hydrophobic tail of xanthate in this figure (such as the surfactant cartoon in Fig. 8 of Fuerstenau and Pradip, 2005, Advances in Colloid and Interface Science, volumes 114-115, pp. 9-26), so that readers can see the orientation of xanthate absorbing on the covellite surface.

Another concern the reviewer has is that authors used spheres to show the crystal growth of covellite. This may be misleading because the covellite formed on the azurite surface after sulfurization should not be in spherical shape.

6.       It would be good to put materials and methods section between introduction and results and discussions so that readers can readily understand the results.

7.       There are two “sulfurization” in the article title.

8.       Ref. 14 and Ref. 18 are duplicated.

Comments on the Quality of English Language

Please see the comments and suggestions for authors.

Author Response

Thank you again for your comments concerning our manuscript entitled “Ammonium–amine co-activation promoting the sulfurization sulfurization of azurite and its effect on xanthate adsorption” (molecules-2617420). Those comments are all valuable and very helpful for revising and improving our paper, and provide the important guiding significance to our studies. We have studied the comments carefully and have made corrections, which we hope to meet with approval. This revised portion are marked in red in the paper. The main corrections in the paper and the respond to your comments are as follows:

Q1: Lines 109-111: The authors attributed the decrease in flotation recovery when overdosing Na2S to the formation of colloidal copper sulfide that prevented xanthate adsorption. However, there might be other possibilities that can cause low flotation recovery in the presence of extra amounts of Na2S. One possible reason was that overdosing Na2S decreased the potential (Eh), which made xanthate adsorption impossible. Without ruling out this possibility, the authors’ statement may be inaccurate and questionable.

Response 1: Based on your comments, it’s not hard to see that you have a deep understanding of sulfurization research of copper oxide. Yes, you are right. The inhibitory effect of excess sodium sulfide is multifaceted. Based on your comments, we have revised this part. The changes have been highlighted in section 3.1.

Q2: What is the washing procedure? What is the difference between unwashed and washed samples shown in Fig. 1a?

Response 2: Thanks for your comments. Based on your comment, we have removed the excess red lines.

Q3: Line 257: What is the definition of “average sizes”? Does it refer to the length of the flakes?

Response 3: Thanks for your comments. This data represents the radial dimension of the sheet product. We have corrected it in manuscript.

Q4: Lines 317-319: Will the sulphuric acid washing change the surface properties of the sulfurized samples for XRD analysis?

Response 4: Thanks for your comments. This method has also been used in previous studies, and XPS has confirmed that the sulphuric acid washing does not affect the chemical state of copper in the resulting copper sulfide. However, the possibility of the effect of the sulphuric acid washing on the crystal phase identification of the product cannot be ruled out. In the follow-up study, we will use other means to further analyze the composition of sulfurized products.

  1. Cai, C. Su, Y. Ma, X. Yu, R. Peng, J. Li, X. Zhang, J. Fang, P. Shen, D. Liu, Role of ammonium sulfate in sulfurization flotation of azurite: Inhibiting the formation of copper sulfide colloid and its mechanism, Int. J. Min. Sci. Technol. 32 (2022) 575-584.
  2. Liu, D. Liu, J. Li, J. Li, Z. Liu, X. Jia, S. Yang, J. Li, S. Ning, Sulfidization mechanism in malachite flotation: A heterogeneous solid-liquid reaction that yields CuxSy phases grown on malachite, Miner. Eng. 154 (2020) 106420.

Q5: In Fig. 10, the authors used short yellow curves to represent xanthate molecules. It would be better if the authors can use a structure that shows the hydrophilic head and hydrophobic tail of xanthate in this figure (such as the surfactant cartoon in Fig. 8 of Fuerstenau and Pradip, 2005, Advances in Colloid and Interface Science, volumes 114-115, pp. 9-26), so that readers can see the orientation of xanthate absorbing on the covellite surface.

Another concern the reviewer has is that authors used spheres to show the crystal growth of covellite. This may be misleading because the covellite formed on the azurite surface after sulfurization should not be in spherical shape.

Response 5: Thanks for your comments. You and other reviewers have been concerned about this problem, and according to your comments, we have revised the hypothesised mechanism model.

Q6: It would be good to put materials and methods section between introduction and results and discussions so that readers can readily understand the results.

Response 6: Thanks for your comments. Based on your comment, we have rearranged the order of the parts.

Q7: There are two “sulfurization” in the article title.

Response 7: Thanks for your comments. The error has been modified.

Q8: Ref. 14 and Ref. 18 are duplicated.

Response 8: Thanks for your comments. The error has been modified.

I hope this revised paper will get your approval.

We deeply appreciate your review of our manuscript, and we look forward to receiving your reply.

Round 2

Reviewer 2 Report

Comments and Suggestions for Authors

All comments were addressed in a suitable form.